# Oral Microbiota Profile in Patients with Anti-Neutrophil Cytoplasmic Antibody–Associated Vasculitis

**DOI:** 10.3390/microorganisms10081572

**Published:** 2022-08-04

**Authors:** Anders Esberg, Linda Johansson, Ewa Berglin, Aladdin J. Mohammad, Andreas P. Jonsson, Johanna Dahlqvist, Bernd Stegmayr, Ingegerd Johansson, Solbritt Rantapää-Dahlqvist

**Affiliations:** 1Department of Odontology, Umeå University, 901 87 Umeå, Sweden; 2Department of Public Health and Clinical Medicine/Rheumatology, Umeå University, 901 85 Umeå, Sweden; 3Department of Clinical Sciences/Rheumatology, Lund University, 221 00 Lund, Sweden; 4Department of Medicine, University of Cambridge, Cambridge CB2 0SP, UK; 5Department of Public Health and Clinical Medicine, Umeå University, 901 85 Umeå, Sweden; 6Department of Medical Sciences/Rheumatology, Uppsala University, 751 85 Uppsala, Sweden; 7Department of Medical Biochemistry and Microbiology, Uppsala University, 752 37 Uppsala, Sweden

**Keywords:** IgG, oral microbiota, vasculitis, caries, periodontal disease, anti-neutrophil cytoplasmatic antibody–associated vasculitis, granulomatosis with polyangiitis (GPA), microscopic polyangiitis (MPA), proteinase 3 (PR3)-ANCA, myeloperoxidase (MPO)-ANCA

## Abstract

Microbiota has been associated with autoimmune diseases, with nasal *Staphylococcus aureus* being implicated in the pathogenesis of anti-neutrophil cytoplasmic antibody–associated vasculitis (AAV). Little is known about the role of oral microbiota in AAV. In this study, levels of IgG antibodies to 53 oral bacterial species/subspecies were screened using immunoblotting in plasma/serum in pre-symptomatic AAV-individuals (*n* = 85), matched controls, and established AAV-patients (*n* = 78). Saliva microbiota from acute-AAV and controls was sequenced from 16s rDNA amplicons. Information on dental status was extracted from a national register. IgG levels against oral bacteria were lower in established AAV versus pre-AAV and controls. Specifically, pre-AAV samples had, compared to controls, a higher abundance of periodontitis-associated species paralleling more signs of periodontitis in established AAV-patients than controls. Saliva microbiota in acute-AAV showed higher within-sample diversity but fewer detectable amplicon-sequence variants and taxa in their core microbiota than controls. Acute-AAV was not associated with increased abundance of periodontal bacteria but species in, e.g., *Arthrospira*, *Staphylococcus*, *Lactobacillus*, and *Scardovia*. In conclusion, the IgG profiles against oral bacteria differed between pre-AAV, established AAV, and controls, and microbiota profiles between acute AAV and controls. The IgG shift from a pre-symptomatic stage to established disease cooccurred with treatment of immunosuppression and/or antibiotics.

## 1. Introduction

The microbiome on and in our body is believed to have evolved along with us and to play a role in health and disease, nutrition modulation, prevention of pathogen invasion, and immune system education [1]. Sequencing technology advances have facilitated culture-independent microbiome analyses showing that dysbiosis may result in excessive immune activation and tissue damage [2]. The microbiome’s importance in disease development and progression has been suggested for several autoimmune conditions, including rheumatoid arthritis (RA) [3], systemic lupus erythematosus [3], inflammatory bowel disease [4], and vasculitis [5]. However, the influence of the microbiome in systemic vasculitis remains unclear, and studies are limited.

Anti-neutrophil cytoplasmic antibody (ANCA)–associated vasculitis (AAV) is a group of diseases characterized by ANCA production, excessive neutrophil activation, and small-medium vessel vasculitis [6]. Mucosal inflammation of the upper and lower respiratory tract is a striking feature in many AAV patients, particularly in those with granulomatosis polyangiitis (GPA), raising suspicions of an infectious trigger [5]. Most microbiome studies have focused on nasal [7,8,9], bronchoalveolar lavage fluid [10,11], and fecal [12] microbiomes, although one report addressed the oral microbiota in children with IgA-vasculitis (Henoch-Schönlein purpura) [13].

In particular, *Staphylococcus aureus* has been the subject of extensive studies in GPA, with *S. aureus* persistence reported in 60–70% of patients [14,15]. Several hypotheses, including induction of neutrophil extracellular traps, molecular mimicry [16], and superantigen production, have been proposed regarding pathogenic mechanisms in breaking the tolerance [17,18]. Whether dysbiosis is causative or an effect of immunosuppressive therapy or vasculitis-related damage is unclear. Although the efficacy of trimethoprim-sulfamethoxazole and its association with a reduced relapse rate [14] support a causal role for *S. aureus* in GPA [18], treatment with trimethoprim-sulfamethoxazole changes the gut microbiome [19], and treatment with immunosuppression alters the nasal microbiome [7,8,20], suggesting a treatment consequence.

Culture-independent sequencing technologies, such as 16S rRNA amplicon sequencing, have confirmed nasal microbiota dysbiosis in GPA patients. The presence of *S. aureus* was confirmed in two studies [7,9], whereas another study showed reductions in *Propionibacterium acnes* and *Staphylococcus epidermidis*, but no differences in *S. aureus* [8]. Cultures of bronchoalveolar lavage samples showed that pathogen isolation was more common in GPA compared with idiopathic pulmonary fibrosis [10]. In another study comparing GPA with sarcoidosis, the alpha-diversity of the lower respiratory tract microbiome was negatively associated with disease activity in AAV. No difference was observed in the microbiome profile between the two diseases [11].

An antagonistic relationship has been suggested between *S. aureus* and *S. epidermidis* [9] with *S. epidermidis* protecting against *S. aureus* by secreting Esp, a serine protease that inhibits *S. aureus* adhesion and activates immune defenses to clear *S. aureus* [21]. Recently, a longitudinal evaluation of the nasal microbiome in GPA (assessed every three months for an average of six years) demonstrated a higher *Staphylococcus* to *Corynebacterium* ratio before flares, with a decline in abundance of *S. epidermidis* and *Propionibacterium acnes* and a rise in *Corynebacterium tuberculostearicum* abundance [8]. That work showed that microbial communities fluctuate with time, disease activity, and treatment. Indirect/direct effects of treatment, such as non-glucocorticoid immunosuppression, were associated with “normalization” of the nasal microbiome independent of disease activity, suggesting that medication may moderate nasal microbes [8]. Furthermore, the intestinal microbiome may modulate the immunosuppressive effects of cytotoxic drugs [22], and oral microbiome profiles correlate with clinical indices and response to therapy in RA [23].

In this study, we analyzed the immune response to oral bacteria and the oral bacterial profile in AAV before symptom onset, in acute AAV, and inactive established disease after treatment in comparison with controls.

## 2. Materials and Methods

### 2.1. Study Participants

This study was performed using plasma, serum, and saliva samples collected from individuals identified before symptom onset (pre-AAV), at the acute manifestation (acute AAV), and during an established stage of AAV (established AAV). As references, non-AAV, clinical controls, and population-based controls were used. All patients with a clinical diagnosis of AAV were classified into disease phenotypes using the European Medicines Agency algorithm [24]. Participant descriptions are presented in Table 1. The Regional Ethics Committee at the University Hospital, Umeå, Sweden approved this study, and all participants gave their written informed consent when donating samples.

### 2.2. Pre-Symptomatic AAV Individuals and Matched Controls Screened for Immunoglobulin (IgG) Antibodies to Oral Bacteria

The process of identifying pre-symptomatic AAV cases has been presented in detail previously [25]. Briefly, the Cause of Death Register and the Swedish National Inpatient Register were used to identify individuals with AAV as a first diagnosis in the discharge summary and/or cause of death between 1987 and 2011 using the codes of the International Classification of Diseases (ICD)-9 (1987–1997; 446.4, 446E) and ICD-10 (1998–2011; M30.1, M31.3, M31.7). Identified personal identity numbers were linked to the registers of five biobanks in Sweden and individuals aged ≥18 years and with a plasma or serum sample donated > 1 month but <10 years before symptom onset were included. Medical records were reviewed to identify the time-point for symptom onset and confirm the AAV diagnosis [24].

In total, 85 pre-symptomatic cases (mean age (standard deviation (SD)) 52.3 (16.7) years; 57.6% women) fulfilled the defined criteria (Table 1). One population-based non-AAV control was matched to each case for sex, age, sampling date, and biobank origin. Of the 85 pre-symptomatic AAV individuals, 11 also had available samples after AAV onset. Of all samples, 80% were serum and 20% plasma, with a similar proportion of each for cases and controls. In the pre-symptomatic AAV cases, 10.6% (*n* = 9) were myeloperoxidase (MPO)-ANCA positive (+), and 24.7% (*n* = 21) were proteinase 3 (PR3)-ANCA+. Of the 11 individuals with a follow-up sample, 27.3% were MPO-ANCA+ and 72.7% PR3-ANCA+. Before sampling, none of the pre-symptomatic individuals or controls had any AAV-related treatment or antibiotics for at least 3 months.

### 2.3. Established AAV Cases Screened for IgG Antibodies to Oral Bacteria

Of the 96 patients diagnosed with established AAV at the Department of Rheumatology and Nephrology, University Hospital, Umeå, Sweden, 78 participated in this study with a plasma sample. Medical records were reviewed to identify the time-point for onset and the AAV diagnosis [24]. The mean (SD) age at sampling was 64.3 (19.1), 52.6% were women, and the disease duration (mean (SD)) was 9.7 (7.1 years). During the disease period, the established AAV patients were treated according to guidelines [26] (Table 1). Thus, during active disease, 88% were treated with pulses of corticosteroids and 81% with cyclophosphamide, and the remainder received azathioprine or methotrexate. In addition, 28% were given antibiotic prophylaxis (trimethoprim-sulfamethoxazole) during active disease. In 10% of the cases with a relapse, rituximab was added. In most cases, maintenance therapy was oral prednisolone (5–12.5 mg daily) plus a cytotoxic drug (e.g., methotrexate/azathioprine/mofetil mycophenolate). 

Information on dental status was retrieved from the Swedish quality register on caries and periodontitis (SKaPa, www.skapareg.se, accessed on 14 December 2021) [27]. Data on the number of teeth, cause of tooth loss, probing pocket depth (PPD), and caries or restoration per tooth (third molars excluded) were available from 2010 to 2020 [28]. For periodontal status, the number of teeth with PPD ≥ 6 mm was calculated as described previously [29], and for caries, the Decayed, Missing, Filled index, which gives the sum of caries-affected tooth surfaces. Information on PPD was available for 61 established AAV cases, and for these, information from the dental visit closest to the AAV diagnosis was retrieved (median difference 3.0 years after AAV diagnosis (quartile limits 0 and 8.5 years). Caries status was available for 59 cases and similarly, the visit closest to the AAV diagnosis was kept (median difference 3 years after AAV diagnosis (quartile limits 0 and 9.0 years). Dental status was also compiled for three controls per case, matched for sex, age, and birth year, and for a group of patients with RA [30] with dental data from the same year as the RA diagnosis (*n* = 557, mean age (SD) 57.7 (15.9) years, 72.4% women, 56.4% ever smoker). Smoking status was classified as being a never or ever (current and former) smoker.

### 2.4. Acute AAV Cases and Controls for Saliva Microbiota Sequencing

Finally, 25 patients with a first acute attack of vasculitis (*n* = 18) or acute relapse (*n* = 7) were recruited (acute AAV) when they were admitted to the University Hospital in Umeå and Uppsala, Sweden, along with 23 healthy volunteers matching the sex and age of the cases. The mean age (SD) of the cases was 60.9 years (18.0), and 60% were women (Table 1). Of the patients, four reported having diabetes, one was a current smoker, and 36.4% were ever smokers, whereas among the controls, 30.4% were ever smokers, and none was a current smoker. Among the acute AAV patients, seven with relapse had been on a continuous low dose of prednisolone, and 10 received corticosteroids on the day of sampling. Furthermore, based on plasma samples, 20% were MPO-ANCA+, and 80% were PR3-ANCA+ (Table 1). In saliva samples, antibodies against MPO-ANCA were found in two MPO-ANCA+ patients, and none had PR3-ANCA antibodies. Of the individuals who had a first acute attack of vasculitis, 9 had received antibiotics between 1–10 days at the primary care and 5 of the individuals with an acute relapse were either on long-term antibiotics or had received it during the previous 2 weeks. Thus, 11 AAV individuals and none of the controls had received antibiotics for the previous 3 months.

Whole stimulated saliva was collected for 3 min into ice-chilled tubes while the participant chewed on a 1-g piece of paraffin wax. Participants had refrained from eating or drinking for 2 h before sampling. All samples were stored at −80 °C until DNA extraction for microbiota analysis. 

### 2.5. Plasma/Serum Antibody Screening

Screening for levels of IgG antibodies to oral bacteria was done by immunoblotting in a checkerboard device, as described previously [31,32]. Briefly, 68 bacterial strains representing (i) oral commensal taxa (e.g., *Corynebacterium matruchotii,*
*Gemella haemolysans,*
*Streptococcus sanguinis)*, (ii) the dysbiotic caries associated biofilm (e.g., *Streptococcus mutans, Lactobacillus gasseri,*
*Lactobacillus vaginalis**),* (iii) the dysbiotic periodontitis associated biofilm (e.g., *Aggregatibacter actinomycetemcomitans, Filifactor alocis, Fusobacterium nucleatum, Porphyromonas gingivalis, Tannerella forsythia)*, or (iv) reported as associated with AAV (*Staphylococcus aureus, Staphylococcus epidermidis*) were grown for 48 h on Columbia-based blood agar, Rogosa, or chocolate agar plates under aerobic or anaerobic conditions at 37 °C (Appendix A). Harvested bacteria were resuspended, washed two times in 50 mM Tris-HCl (pH 7.5) with 150 mM NaCl (1× Tris-buffered saline [TBS]), before being stored at −80 °C in 500-µL aliquots at an optical density of 1.0 at 600 nm. Nitrocellulose membranes (Amersham™ Protran^®^ GE10600003, Merck, Solna, Sweden) were pre-wet with ultrapure water, soaked in TBS (10 min), before applied in a Miniblotter device. Bacterial suspensions (150 µL), positive controls (Protein A, P6031, Merck) at 0.005–1 µg/mL, and negative controls (TBS) were loaded into the lanes of the Miniblotter (Miniblotter 45MN, Interchime, Montlucon Cedex, France) and incubated overnight at 4 °C. The liquids were removed from the lanes, washed with TBS with 0.1% Tween 20 (TBS-T), and blocked in TBS-T containing a 5% blocking reagent (ECL™ Advance Blocking Reagent, GERPN418, Merck, Solna, Sweden). Plasma/serum diluted 1:500 in TBS was applied perpendicular to the bacteria and incubated for 1 h at room temperature. The liquids were removed, and the membrane rinsed 3 × 1 min in TBS-T before treatment with 0.3% H_2_O_2_ (H1009, Merck) for 10 min to reduce endogenous bacterial peroxidase activity. Then the membrane was washed 1 × 15 min and 3 × 5 min using TBS-T followed by incubation with an anti-human-IgG-Fab peroxidase-labeled secondary antibody (Anti-Human-IgG-Fab, A0293, Merck) in TBS-T with 5% blocking powder for 1 h at room temperature under slow rotation. Finally, the membrane was washed (1 × 15 min and 3 × 5 min) using TBS-T before signal development (ECL™ Prime Western Blotting Detection Reagent, GERPN2236, Merck). ChemiDoc^TM^ XRS+ System (Bio-Rad, Solna, Sweden) was used for signal detection.

In addition, serum/plasma samples from pre-AAV individuals were screened for IgG class ANCA using high-sensitivity ELISA (ORGENTEC Diagnostika, Mainz, Germany) with the cut-off for positivity set by the manufacturer as ≥1. Furthermore, the pre-AAV serum/plasma samples and acute AAV saliva samples were analyzed using second-generation (capture-based) PR3-ANCA and MPO-ANCA ELISAs (SVAR Life Science, Malmö, Sweden) with cut-offs at 7 and 8 IU/mL, respectively, as previously described [25]. Saliva samples were diluted 10 times more than serum/plasma samples.

### 2.6. Saliva Microbiota Sequencing

DNA was extracted from saliva using the GenElute Bacterial genomic DNA kit (Sigma-Aldrich Co, Stockholm, Sweden) from 400 µL of saliva. In short, saliva was thawed on ice, centrifuged (5 min at 13,000× *g* rpm), lysed in a buffer with mutanolysin and lysozyme, followed by RNase and Proteinase K treatment. DNA was purified, washed, and eluted at room temperature in 150 µL elution buffer. DNA quality was assessed using a NanoDrop 1000 Spectrophotometer (Thermo Fisher Scientific, Uppsala, Sweden) and quantified using the Qubit 4 Fluorometer (Invitrogen, Thermo Fisher Scientific, Waltham, MA, USA). The DNA concentration ranged from 17.5 to 26.8 ng/µL. The same DNA extraction method was applied to negative (Milli-Q Ultrapure water) and positive controls (mock mixture of 14 oral bacterial species).

Bacterial V3-V4 16S rDNA amplicons were generated by PCR amplification using forward primers 341F (ACGGGAGGCAGCAG) and the reverse primers 806R (GGACTACHVGGGTWTCTAAT) from saliva, and positive and negative control extracted DNA as described by Caporaso et al. [33]. An equal amount of DNA was applied to each PCR reaction (50 ng), and equal amounts of amplicon libraries were pooled before purified using AMPure XP beads (Beckman Coulter, Stockholm, Sweden). All samples were analyzed in one run by Illumina Miseq sequencing 2 × 300 bp kit (Illumina, Stockholm, Sweden) at the Swedish Defense Research Agency research facility in Umeå, Sweden, including a 5% PhiX and 12.5 pM amplicon library. Acquired sequences were demultiplexed, the pair-end reads fused, primers, chimeric and ambiguous sequences, and PhiX removed, and amplicon sequence variants (ASV) identified using the open-source software package DADA2 in the QIIME2 (https://qiime2.org, accessed on 10 December 2021) [34,35]. ASVs were taxonomically classified against the expanded Human Oral Microbiome Database (eHOMD, http://www.homd.org, accessed on 10 December 2021) [36]. ASVs with ≥2 reads and 98.5% identity with a named species or unnamed phylotype in eHOMD were retained, and those with the same Human Microbial Taxon (HMT) ID number were aggregated. The mock (positive control) that was used contained a mixture of *Actinomyces odontolyticus*, *Bifidobacterium longum*, *Bifidobacterium dentium*, *Corynebacterium matruchotii*, *Gemella haemolysans*, *Haemophilus parainfluenzae*, *Lactobacillus fermentum*, *Lactobacillus vaginalis*, *Porphyromonas*
*gingivalis*, *Rothia mucilaginosa*, *Streptococcus intermedius*, *Streptococcus*
*mutans*, *Streptococcus*
*sanguinis*, *Streptococcus parasanguinis*. All 14 mock-included species were detected for each batch, and no reads were generated for the negative controls (ultrapure water) using the same bioinformatic criteria as for the samples.

In addition, targeted detection of quantitative PCR analyses was performed in a QuantStudio 6 system (Applied Biosystems by Life Technologies, Carlsbad, CA, USA) using TaqMan Universal Master Mix (Applied Biosystems) and TaqMan kit predesigned to detect and quantify *Staphylococcus* in saliva [37]. All samples were run in duplicate and quantified against a standard curve from 1 to 1 × 10^−5^ ng/µL DNA isolated from the *Staphylococcus aureus* CCUG64138 strain.

### 2.7. Statistical Analyses

SPSS (IBM Corp. version 27.0) and PAST 4 software packages were used for descriptive statistics, including means and medians, standard deviations (SD), 95% confidence interval (95% CI), and proportions (%). Group differences between continuous variables were tested using the Mann–Whitney U test and categorical variables using the chi-square test (χ^2^) or Fisher’s exact test. Differences between the mean (95% CI) of the number of PPD > 6 mm teeth and DMFS were evaluated using general linear modeling (GLM) with adjusted for sex, age, and smoking and sensitivity analyses for birth year and the time difference between dental recordings and AAV diagnosis.

Binary logistic regression was performed with subgroups of interest as the dependent variable and antibody response to each of the 68 bacterial strains, including sex, age, and sample type (plasma/serum) as covariates. Results are presented as bar graphs based on beta values (β) and standard errors (SE).

Microbiota diversity was evaluated for alpha-diversity using the Shannon diversity index (which considers abundance and evenness), evenness index (which evaluates evenness), and Faith PD index (a measure of biodiversity based on phylogeny). Beta-diversity was also evaluated by Bray–Curtis dissimilarity (based on species abundances) and the Jaccard index (based on presence), Unweighted UniFrac (based on phylogenetic similarities of presence), and Weighted UniFrac (based on phylogenetic similarities of abundances) using QIIME2 [35]. All tests were two-sided, and *p* < 0.05 was considered significant after controlling for multiple testing; adjusted *p* values are presented as the false discovery rate–derived *q* value.

Multivariate analyses included unsupervised principal component analysis (PCA) for group separation and supervised orthogonal projection to latent structures-discriminant analysis (OPLS-DA) to evaluate binary classification. SIMCA P+ version 17.0 (Sartorius Stedim Data Analytics AB, Malmö, Sweden) was used. PCA analysis evaluated participants’ microbiota patterns using the identified genus or species abundance and presence. OPLS-DA analysis was used to identify genera and species associated with being a control or vasculitis patient or clinical treatments related to AAV. All variables were log-transformed using the SIMCA function “auto transform selected variables as appropriate” and scaled to unit variance when needed. K-fold cross-validation was performed by systematically removing every seventh observation and predicting the remaining observations (Q^2^-values and analysis of variance [ANOVA] of cross-validated residuals. The results were displayed in two-dimensional score loading plots projecting component 1 with the maximal separation of the observations and orthogonal component (to[1]) representing within-group variation. Multivariate partial least squares regression modeling estimates the explanatory and predictive power of many (and even co-varying) x-variables when modeled simultaneously against an outcome(s) (y-variable(s)). Variable importance in the projection (VIP-values) reflects the significance of each x-variable in explaining the variation in y. VIP values are presented for the predictive components only, with VIP >1 considered significant.

For high-dimensional class comparisons of the microbiota of controls and vasculitis patients linear discriminant analysis (LDA) effect size (LEfSe) method was used [38].

GLM was used to evaluate unadjusted PCA loading scores (t[1] or to[1]) in subgroups, or adjusted for potential confounding factors, i.e., sex, age, ever/never smoking, diabetes, hypertension, antibiotic treatment, and MPO and PR3 antibody profile.

## 3. Results

### 3.1. Immune Response to Oral Bacteria in Pre-Symptomatic versus Established AAV Individuals

To compare adaptive immune responses to oral bacteria in pre-symptomatic (pre-AAV, *n* = 85) versus established AAV (*n* = 78) stages and in controls (*n* = 85), IgG levels against 53 oral bacterial species/subspecies in 27 genera were screened by immunoblotting. Total IgG levels were similar between pre-AAV and control samples (*p* = 0.105) but significantly lower in established AAV against both pre-AAV and control samples (*p* = 0.0016 and *p* = 0.021, respectively). Nevertheless, the global IgG pattern relative to the 53 species/subspecies differed significantly between the control, pre-AAV, and established AAV subgroups (*p* = 0.0003, one-way permutational multivariate ANOVA). Hence, compared with the controls, pre-AAV individuals had significantly higher levels of IgGs against *Actinomyces massiliensis*, *Fusobacterium nucleatum subsp. polymorphum*, *Filifactor alocis*, *Fusobacterium nucleatum vincentii*, *Campylobacter rectus concave*, *Fusobacterium nucleatum subsp. animalis*, *Prevotella pleuritidis*, *Haemophilus parainfluenzae*, *Prevotella intermedia*, and *Prevotella nigrescens* (all *p* ≤ 0.001)) (Figure 1a), whereas samples from controls had higher levels of *Streptococcus oralis* and *Campylobacter gracilis*. Compared with pre-AAV samples, established AAV samples had significantly lower levels of IgGs against 19 species (*p* ≤ 0.001; Figure 1b): *Tannerella forsythia*, *Prevotella nigrescens*, *Aggregatibacter actinomycetemcomitans*, *Prevotella pleuritidis*, *Streptococcus oralis*, *Dialister invisus*, *Streptococcus sanguinis*, *Campylobacter rectus concave*, *Campylobacter rectus convex*, *Lautropia mirabilis*, *Actinomyces massiliensis*, *Veillonella rogosae*, *Fusobacterium nucleatum subsp. polymorphum*, *Fusobacterium periodonticum*, *Filifactor alocis*, *Bifidobacterium longum*, *Selenomonas noxia*, *Fusobacterium nucleatum subsp. nucleatum*, and *Veillonella dispar*. Compared with controls, patients with established AAV had lower levels of IgGs to *Lachnoanaerobaculum saburreum comb. Nov.*, *Streptococcus gordonii*, *Lautropia mirabilis*, *Bifidobacterium dentium*, *Streptococcus oralis*, *Streptococcus sanguinis*, *Streptococcus mitis*, *Campylobacter gracilis*, *Rothia mucilaginosa*, *Veillonella dispar*, and *Bifidobacterium longum* (Figure 1c). Thus, the abundances of *Streptococcus oralis* and *Campylobacter gracilis* were increased in controls compared with both pre-and established AAV (Figure 1a,c). Within pre-AAV samples, ANCA (PR3+ or MPO+) status was not associated with a changed total IgG amount or IgG pattern (*p* > 0.05).

Eleven pre-AAV individuals had follow-up samples as established AAV patients, allowing for a longitudinal evaluation of the total and specific IgG during disease progression. In line with the cross-sectional findings, total IgG levels had declined in established AAV compared with the pre-symptomatic stage (*p* = 0.0028), as had levels for 22 bacterial species (*p* ≤ 0.010), of which 10 overlapped with those identified in the cross-sectional comparison: *Aggregatibacter actinomycetemcomitans*, *Bifidobacterium longum*, *Filifactor alocis*, *Fusobacterium nucleatum subsp nucleatum*, *Fusobacterium periodonticum*, *Lautropia mirabilis*, *Prevotella pleuritidis*, *Streptococcus sanguinis*, *Streptococcus oralis*, and *Tannerella forsythia*. Figure 2 shows four examples.

### 3.2. Dental Status in Established AAV

Given elevated IgGs to several periodontitis-associated oral species in pre-AAV individuals but decreased levels in established AAV, we compared periodontitis severity between established AAV and matched control samples using deep PPD (≥6 mm) as a proxy. After adjustment for smoking, patients with established AAV had three times more deep pockets than controls matched for gender, age, and smoking (mean (95% CI) 1.3 (0.8, 1.8) versus 3.9 (3.1, 4.8), *p* < 0.001). Additional adjustment for birth year did not affect the numbers, whereas adjustment for time difference between dental examination and AAV diagnosis attenuated the difference somewhat, but it remained statistically significant (*p* = 0.020). Restricting the analysis to cases with the dental examination the same year as the AAV diagnosis (*n* = 20) and their matched controls yielded similar mean values as for the whole group, i.e., 1.3 versus 3.5 PPD 6 mm, *p* = 0.032. For comparison, we compared the number of deep pockets in the established AAV group with values in early RA patients, as RA has been reported as associated with deteriorated periodontal status. After adjustment for smoking, the RA group had four times more deep pockets than equally matched controls (mean (95% CI) 1.3 (1.2, 1.5) versus 4.8 (4.5, 5.0), *p* < 0.001). Findings for caries-affected tooth surfaces did not differ between individuals with established AAV and matched controls (mean Decayed, Missing, Filled index (95% CI) 52 (46, 57) versus 49 (40, 59), *p* = 0.735).

### 3.3. Illumina Amplicon Sequencing of Saliva Microbiota in Acute AAV Samples

Given the altered antibody profile and increased prevalence of periodontitis in AAV patients and that no oral samples were available in the biobanks, we recruited a group of individuals with acute AAV for characterization of the oral microbiota at disease onset. Forty-eight saliva samples (25 from acute AAV patients and 23 from controls) were analyzed for microbiota composition by V3-V4 16S rRNA gene amplicon sequencing. After merging, trimming, denoising, and removal of potential chimeric sequences, there were 13,028,507 reads retained. These reads generated 2,627,201 paired end-fusion reads, which corresponded to 2623 ASVs. Of these ASVs, 6.2% (*n* = 162) had no match in *e*HOMD, and 11.6% (*n* = 305) did not meet the criterion of 98.5% identity with a sequence in the *e*HOMD 16S rRNA gene database, but 82.2% (*n* = 2156 ASVs) matched a sequence at 98.5% (or higher) identity. These 2156 ASVs represented 376 species in 11 phyla and 96 genera with >2 reads. Represented phyla were, in descending order: Firmicutes (50.3%), Bacteroidetes (20.3%), Proteobacteria (13.0%), Actinobacteria (11.7%), Fusobacteria (3.52%), Saccharibacteria (TM7) (0.77%), Cyanobacteria (0.28%), Spirochaetes (0.09%), Absconditabacteria (SR1) (0.02%), Gracilibacteria (GN02) (0.01%), and Synergistetes (0.005%). The 15 most abundant genera were *Streptococcus* (28.0%), *Prevotella* (14.4%), *Veillonella* (13.5%), *Rothia* (6.53%), *Haemophilus* (6.22%), *Neisseria* (5.78%), *Porphyromonas* (3.56%), *Gemella* (2.30%), *Fusobacterium* (2.24%), *Granulicatella* (1.84%), *Schaalia* (1.70%), *Corynebacterium* (1.42%), *Leptotrichia* (1.28%), *Alloprevotella* (1.04%), and *Actinomyces* (1.00%) (Appendix A).

### 3.4. Microbiota Diversity Characterization in Acute AAV versus Controls

Compared with controls, individuals with acute AAV had significantly higher within-sample diversity (beta diversity) in their saliva microbiota regardless of whether the comparisons were between quantitative (Bray–Curtis distance matrix, *q* = 0.001; Weighted UniFrac distance matrix, *q* = 0.018) or qualitative measures (Jaccard distance matrix, *q* = 0.001; Unweighted UniFrac distance matrix, *q* = 0.006) (Figure 3a–d). In contrast, they had significantly less richness (Shannon index, *q* = 0.034) and evenness (Evenness index, *q* = 0.042) than controls (Figure 3e,f) and tended to have fewer detectable ASVs (*q* = 0.101) (Figure 3g,h) in saliva, but phylogenetically their saliva microbiota did not differ from controls (Faith index, *q* = 0.942) (Figure 3i).

To understand the nature of the beta-diversity in acute AAV samples, we evaluated the number of shared taxa (“core” microbiota) by sliding cut-off levels for the proportions of shared species. The analyses were done separately for the acute AAV and control groups. When all participants were expected to harbor a species, control samples had 16 species in the core microbiota compared with two species in samples from the acute AAV group (∆-14; Figure 4). The “core” microbiota consistently contained fewer species in the acute AAV samples than in control samples until the requirement was that 10% of the respective group members should harbor the species. At lower levels, the diversity was higher in the acute AAV group than in controls.

### 3.5. LEfSe Identified Differences in Saliva Microbiota in Acute AAV versus Control Samples

LEfSe analysis revealed that compared with matched controls, samples from acute AAV patients had lower abundances of several genera, most notably *Haemophilus*, *Fusobacterium*, *Alloprevotella*, *Schaalia*, and *Leptotrichia* (Figure 5a,b, LDA score > 2.0, *p* < 0.05). Additionally, they had higher relative abundances of *Arthrospira*, *Cariobacterium*, *Lactobacillus, Ruminococcaceae* G-1, and *Staphylococcus* (Appendix A). These results were confirmed in non-parametric, univariate analyses.

Illumina sequencing identified the *Staphylococcus* genus in 24% (*n* = 6) of the acute AAV cases but none of the controls (*p* = 0.023) (Appendix A). Sequencing detected *Staphylococcus aureus* in two AAV cases but none of the controls (*p* > 0.05). Sensitivity analysis (quantitative PCR) also detected *Staphylococcus aureus* in 6 AAV cases and in 2 controls (*p* > 0.05). Thus, species in the *Staphylococcus* genus tended to be more prevalent in saliva from acute AAV than from controls, whereas *Staphylococcus aureus* did not.

### 3.6. Data-Driven Profiling of Saliva Microbiota in Acute AAV versus Controls

Multivariate PCA modeling of abundance and prevalence of taxa at the genus level (R^2^ = 29%, Q^2^ = 16%) indicated that a fraction of acute AAV patients clustered distinctly apart from the controls and other AAVs groups (Figure 6a). GLM using the PCA loading scores confirmed a significant difference between the sample groups for component 1 (*p*_t[1]_ = 0.00021) but not component 2 (*p*_t[2]_ = 0.051). Sensitivity analysis, adjusted for sex, age, and smoking, did not alter the significant difference for component 1 (*p* = 0.000041). Subsequent OPLS-DA analysis with AAV status as the dependent variable and the PCA set of independent variables generated a stable model that explained 68% and predicted 35% of the sample variation (*p* = 0.00091 in PLS CV-ANOVA) (Figure 6c). Important genera (VIP > 1.2) for the separation by component 1 that were increased in abundance and/or detection in acute AAV samples were *Arthrospira*, *Fretibacterium*, *Lactobacillus*, *Scardovia, Staphylococcus* and *Veillonellaceae* [G-1]), while reduced abundance and/or prevalence of 31 genera (including *Aggregatibacter*, *Alloprevotella*, *Bergeyella*, *Butyrivibrio*, *Catonella*, and *Fusobacterium*) (Appendix A).

Repeating the PCA analysis with abundance and prevalence of taxa at the species level yielded results similar to those for the genus level, i.e., acute AAV samples separated from the controls in component 1 (*p*_t[1]_ = 0.00001) but not component 2 (*p*_t[2]_ = 0.221) in a modestly strong model (R^2^ = 20%, Q^2^ = 7%) (Figure 6b). Additionally, the OPLS-DA model comparing the acute AAV cases and controls was strong, with 88% explained and 25% predicted of the model variation (*p* = 0.015 in PLS CV-ANOVA) (Figure 6d).

The model recognized 15 enriched species (VIP > 1.2) in the acute AAV samples; top candidates were Campylobacter gracilis, Arthrospira platensis, Lactobacillus fermentum, Treponema socranskii, Lactobacillus paracasei, Scardovia wiggsiae, Oribacterium sp. HMT 078, and Streptococcus mutans. In line with both LEfSe and OPLS-DA results for genera, vasculitis patient samples displayed a decreased abundance of a large number of species (*n* = 76) (VIP > 1.2) compared with control samples. The top candidates were Fusobacterium periodonticum, Solobacterium moorei, Peptostreptococcaceae [XI][G-1] sulci, Leptotrichia sp. HMT 417, Stomatobaculum sp. HMT 097, Haemophilus parainfluenzae, Saccharibacteria (TM7) [G-1] bacterium HMT 352, and Mogibacterium diversum (Appendix A).

### 3.7. Acute AAV Subgroup Characterization

The PCA score plot (Figure 6a) indicated that the acute AAV cases represented at least two subgroups. An OPLS-DA model restricted to the acute AAV cases explained 95% and predicted 80% of the bacteria profile variation among the acute AAV cases (*p* = 1.2 × 10^−6^) (Figure 7a). In this model, antibiotic treatment emerged as a significantly influential variable for clustering the acute AAV patients (VIP = 1.27). Sex, age, diabetes, hypertension, GPA vs. MPA AAV, PR3-ANCA or MPO-ANCA antibody profile, or glucocorticoid treatment were all non-influential (Figure 7b). The results were consistent between the use of genera or species for the OPLS-DA analysis.

To follow up on the systematic impact of antibiotic exposure in the acute AAV cases we rerun the model from Figure 7 and included the 11 cases who did not have any antibiotic exposure for the previous 3 months. This model confirmed enrichment of *Staphylococcus aureus* in the acute AAV cases (VIP 2.0) compared with the controls, as well as several caries-associated species with a VIP-value > 1.2 (7 species in *Lactobacillus*, *Streptococcus mutans, Scardovia wiggsiae*, 2 species in *Bifidobacterium*, *Prevotella denticola* and *Veillonellaceae* [G-1]). However, the 11 acute AAV cases that did not have antibiotic treatment for the previous 3 months also had higher abundances of several species that have been reported as being associated with periodontitis. i.e., *Aggregatibacter actinomycetemcomitans*, 2 species in *Actinomyces and Fusobacterium*, *Dialister invisus*, *Prevotella nigrescens, Tannerella forsythia,* and 5 species in Treponema compared with controls. A full list of taxa with VIP-values >1.2 is presented in Appendix A.

## 4. Discussion

In this study, IgG responses to oral bacteria and saliva microbiome profiles were compared in individuals with AAV in three different stages of the disease. Although the total IgG response to the tested panel of oral bacteria did not differ between controls and pre-symptomatic AAV, the profiles did, as did the profiles and total IgG levels in cases with established AAV. Furthermore, the saliva bacteria profile in acute AAV cases differed distinctly from controls.

The IgG profiles of 53 oral bacterial species/subspecies representing commensal and opportunistic taxa differed distinctly between the controls and pre- and established AAV participants, respectively. Thus, IgG levels in several species that have been associated with periodontitis, such as *F. alocis*, *T. forsythia*, and *A. actinomycetemcomitans*, and species in *Prevotella and*
*Fusobacterium* were elevated in pre-AAV individuals but reduced in established AAV patients [39,40,41]. These species have previously been shown to be more prevalent in other autoimmune diseases [13,28,42,43], and based on this, a link has long been hypothesized between periodontitis and chronic inflammatory disease, such as RA [44]. In support of these findings, established AAV patients had more teeth with deep periodontal pockets than controls, but still with less pronounced signs than patients with established RA. In fact, oral manifestations, such as gingival inflammation (strawberry gingivitis), ulcerations, and tooth loss, have been reported to be present in 6–13% of GPA patients [45,46,47]. Additionally, an unexpected, but noteworthy, finding was that the established AAV status was not only characterized by lower serum IgG levels to periodontitis associated bacteria but also to commensal bacteria in the oral core microbiota, i.e., species commonly found in all or almost all subjects, such as *S. mitis*, *S. sanguinis*, *S. oralis*, and *R. mucilaginosa*. Combined with the contrasting finding of enrichment of periodontitis-associated species in pre-AAV the present findings support a hypothesis of different dysbiotic oral microbiomes in the pre- and established AAV stages due to host traits, disease, or treatment-related exposure.

Unfortunately, the blood biobanks did not have samples suitable for the characterization of the oral microbiota. For this reason, we identified a group of consecutively included AAV patients at the acute stage of the disease and a few with an acute relapse and collected saliva for microbiota characterization. The most prominent findings among all 25 acute AAV patients, i.e., half of the group having antibiotic treatment at sampling and at least for days, were lower saliva microbiota richness (alpha-diversity), but higher beta-diversity, than controls. Here, a combination of LEfSe analysis and multivariate modeling suggested increased detection prevalence of species in the genera *Arthrospira*, *Cardiobacterium*, *Lactobacillus*, *Ruminococcaceae* G-1, *Staphylococcus, Fretibacterium*, *Veillonellaceae* [G-1], and *Scardovia*. When the analysis was restricted to the 11 acute AAV cases who did not have antibiotic exposure for at least 3 months higher abundance was confirmed for these (except *Cardiobacterium*, and *Ruminococcaceae* G-1) and more genera, such as *Streptococcus* and *Treponema*. A higher abundance of *Streptococcus* has been reported in active GPA [7], which was confirmed in the non-antibiotic acute AAV group, but antibiotic treatment appeared to abolish the enrichment despite *Streptococcus* still being the most abundant genus in the saliva of all participants (28%).

Among the 11 non-antibiotics exposed acute AAV cases, several species known to be associated with periodontitis, such as *Aggregatibacter actinomycetemcomitans*, *Tannerella forsythia,* and species in *Fusobacterium* and *Treponema* were enriched compared to the controls, as were several key actors in the development of dental caries, namely *Streptococcus mutans*, *Scardovia wiggsiae,* bifidobacteria, and lactobacilli. When including the antibiotic-treated acute AAV cases the enrichment of caries-associated species was still seen but the enrichment of periodontitis-associated species was nullified. We lacked information on dental status in the acute AAV group, but the higher abundance of periodontitis-associated species was in line with more signs of periodontitis in established AAV cases, whereas no difference was seen in caries experience compared with controls. Information from other studies is lacking. The findings in sera/plasma of the microbiota from the different stages of AAV did not parallel the microbiota findings in saliva from acute AAV cases. This may reflect that the samples were not from the same individuals and that the acute AAV group was comparably small.

The IgG response to the test panel of oral bacteria species was significantly lower in established AAV samples than in pre-AAV and control samples. This finding may suggest that either the transformation from a pre-symptomatic stage to an established disease or the immunosuppressive treatment given to AAV patients induces an overall lowered IgG response to microorganisms, including oral bacteria. Patients in the established AAV group had been treated extensively during the active course of the disease, and at the blood sampling, 58% were still on oral glucocorticoids and 89% on cytotoxic drugs. Furthermore, 28% had been treated prophylactically with trimethoprim-sulfamethoxazole (an antibiotic mix) during the active course of the disease. Along this line, Kronbichler et al. reported that treatment with trimethoprim-sulfamethoxazole reduced bacteria diversity and disease relapses [48], and Rhee et al. showed that non-glucocorticoid immunosuppression normalized bacterial and fungal composition in the noses of GPA patients and reduced bacteria diversity in AAV patients compared with controls [8]. Whether the prophylactical treatment with antibiotics or the immune-suppressive treatment contributes to the lower IgG responses to oral bacteria in established AAV is not possible to distinguish in our study partly due to the low incidence of the disease (limited number of available cases) and partly due to that many patients had a cocktail of medications.

We have recently reported that about 37% of symptom-free pre-AAV individuals were ANCA (PR3- or MPO-ANCA) positive in samples collected between 1 month and 10 years before AAV debut [25]. Thus, ANCA positivity is suggested as an early marker of disease development similar to what was reported for antibodies against citrullinated proteins (ACPA) in pre-RA [49]. Further, activation of the complement system as an early event in the progression of AAV was also reported by us [50]. The enhanced immune response to oral bacteria in pre-AAV versus controls may be part of a general immunological disturbance before clinical manifestation of AAV or potentially reflect an ongoing periodontal inflammatory process triggering the development of autoimmunity or that the inflammatory progression in pre-AAV affects the bacterial profile. However, analyses on stratification for the presence of ANCA as an early sign of disease progression did not support any of the suggested pathways.

Previous studies have shown that 60–70% of PR3-ANCA–positive GPA patients are chronic carriers of *Staphylococcus aureus* in the nose, compared with 20–30% of unaffected individuals [14,51]. Accordingly, *S. aureus* has been suggested to be linked to AAV disease activity and relapse. Furthermore, antibacterial treatment has been reported to reduce these events [14,17]. We found that 24% of the acute AAV cases had detectable *Staphylococcus* in saliva versus none among the controls (*p* < 0.05), with higher abundances in non-antibiotic treated acute AAV cases versus controls but not when antibiotic-treated cases were included. The latter finding agrees with findings by Rhee et al., who reported no difference in the relative abundance of *S. aureus* between nose samples from GPA cases and controls [8]. However, as with the IgG profiling, the microbiota findings, including a higher prevalence of *Staphylococcus,* do not allow for the distinction of causal associations and potential effects of acute-phase or long-term treatments. Thus, though saliva was collected in acute cases of AAV the majority (19/25) had not yet entered AAV treatment, 14 of the 25 patients had received antibiotics within the previous 3 months. i.e., 5 with acute AAV who had a relapse with previous treatments, and 9 on suspicion of an infection.

The uncertainty of the role of treatment versus the bacteria per se and the finding that the controls formed a narrow cluster, whereas the acute AAV cases were scattered into at least two subgroups in the score plot from the microbiota-based PCA prompted further evaluations. As the scattering among the AAV patients suggested that underlying driving forces beyond AAV status were influential, follow-up OPLS-DA was run among the cases only. This revealed treatment with antibiotics as a significant factor for the sub-group separation and possibly glucocorticoid exposure as a candidate factor (though not statistically significant), whereas all other clinical data (autoantibodies, smoking habits) or comorbidity (diabetes, hypertension) were non-influential for the PCA subgroup separation in acute AAV. Thus, the results by Rhee et al. of glucocorticoid effects on nasal microbiota [8] were not fully supported but our limited group size calls for further evaluations. Taken together, our results emphasize that the effects of acute-phase and long-term treatments on the microbiota and analyses carried out on individuals undergoing treatment should be interpreted with caution.

The strength of the present study was the availability of samples from biobanks, which allowed the identification of individuals with pre-AAV and patients with established AAV, together with samples from controls. The biobank search yielded a unique identification of pre-AAV individuals not previously published and a comparably large group of established AAV patients. Furthermore, in this study, we could follow a subgroup of 11 AAV patients from a pre-symptomatic to an established stage of the disease. A weakness is the limited sample size, reducing the power for sensitivity analyses of treatments, such as antibiotics, corticosteroids and/or cytotoxic drugs. An additional weakness is that the assessment of periodontitis was based on pocket depth alone; however, clinical attachment loss or bone loss are not yet systematically reported to the national register on dental status in Sweden. Another potential weakness is the limitation of the V3-V4 fragments in distinguishing phylogenetically close bacterial species, such as *Streptococcus* and *Lactobacillus*. To limit the effect of misclassification, we requested 98.5% similarity with the matched database sequences, accepted only species represented by at least two sequences, and performed analyses at both the genus and species levels. Furthermore, the 16S rDNA amplicon sequencing method is restricted to bacterial identification and does not identify fungi, protozoa, and viruses. Future studies should aim to target the potential causal roles of bacteria colonizing the oral cavity in AAV or other forms of vasculitis because many oral species are sources for gut-, pharyngeal-, and airway-colonizing species.

## 5. Conclusions

The conclusions from the present study are that the immune response to oral bacteria differed significantly between pre-AAV, established AAV, and controls, but with different dysbiosis profiles in the two AAV states. It was also concluded that periodontitis manifestations were more severe in AAV patients than in controls but did not reach the levels seen in RA patients. Whether these differences reflect variation in the microbiota per se or the immune response cannot be distinguished from the present results. Similarly, it was concluded that saliva microbiota communities differed between AAV patients with acute disease and controls but conclusions on causality cannot be drawn. Furthermore, our results suggest that immunosuppressive treatment after manifested diagnosis reduces the bacterial overload identified at the pre-symptomatic stage to established disease. The increased immune response to oral microbiota may indicate a clinically not manifest AAV already years before symptom onset. However, this needs to be evaluated in future human studies or experimental settings.

## Figures and Tables

**Figure 1 microorganisms-10-01572-f001:**
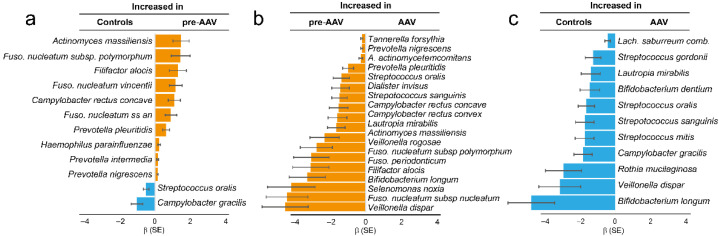
Levels of IgGs against oral bacteria in samples from controls and patients with pre-symptomatic AAV (pre-AAV) and established AAV (AAV). Species with significant differences (*p* ≤ 0.001) are displayed. Checkerboard immunoblotting was used to detect IgG levels against 53 oral bacterial species/subspecies. Signals were evaluated by logistic regression adjusted for sex, age, and sample type. The bars show beta values (β) and standard error (SE) for the subgroups. Blue indicates increased levels in controls, and orange indicates increased levels in pre-AAV samples.

**Figure 2 microorganisms-10-01572-f002:**
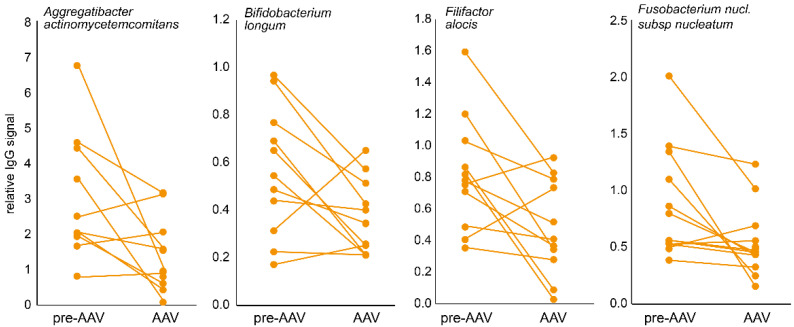
Typical longitudinal IgG changes in samples from pre-AAV stage individuals and when established AAV patients. Dot plots connected with a line illustrate the individual change from the pre-AAV to the established AAV stage for four bacterial species.

**Figure 3 microorganisms-10-01572-f003:**
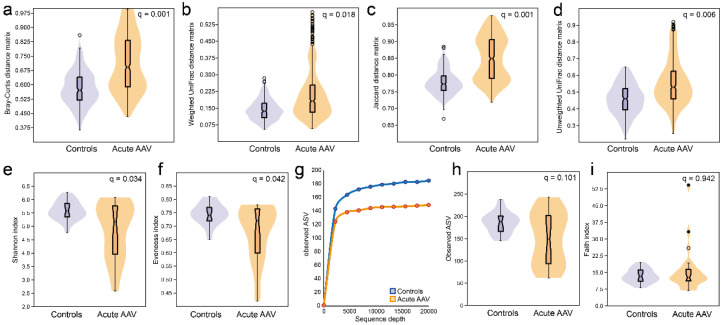
Diversity measures of saliva microbiota in acute AAV and control individuals. Violin plots illustrating beta-diversity based on (**a**) quantitative phylogeny measures by the Bray–Curtis distance matrix (*q* = 0.001) or (**b**) Weighted UniFrac distance matrix (*q* = 0.018) or qualitative measures (**c**) without phylogeny by the Jaccard distance matrix (*q* = 0.001) or (**d**) with phylogeny by the Unweighted UniFrac distance matrix (*q* = 0.006). Alpha-diversities for (**e**) richness and evenness (Shannon index, *q* = 0.034), (**f**) evenness (Evenness index, *q* = 0.042), (**g**) the number of ASVs at different sequencing depths, (**h**) violin plots with box plots for the total number of ASVs at 20,000-read sequencing depth, and (**i**) phylogenetically (Faith index, *q* = 0.942).

**Figure 4 microorganisms-10-01572-f004:**
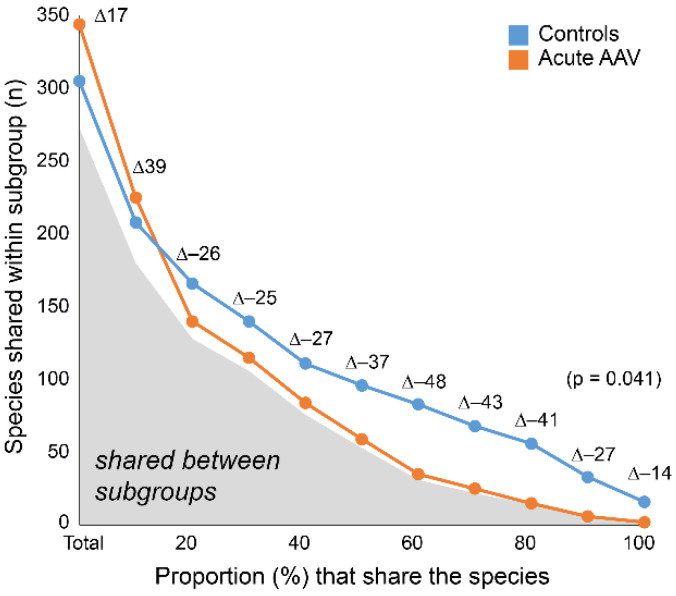
Shared species within and between subgroups. The number of species shared within acute AAV and controls by increasing sharing requirement. Gray area under the curve indicates the number of shared species between subgroups. Delta (∆) values reflect the difference in counts between acute AAV and control participants.

**Figure 5 microorganisms-10-01572-f005:**
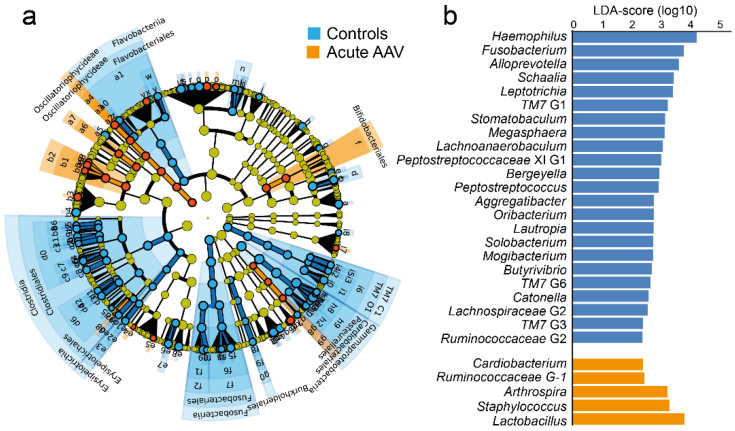
LEfSe analysis of saliva microbiota from controls and acute AAV patients. (**a**) Cladogram, based on phylum, class, order, family, genus, and species abundances in control and acute AAV samples. Indicated taxa differ between the two groups (LDA score > 2.0, *p* < 0.05). (**b**) Bar graphs showing LDA scores for genus-level differences between the control and acute AAV groups. LDA scores are given on a log10 scale.

**Figure 6 microorganisms-10-01572-f006:**
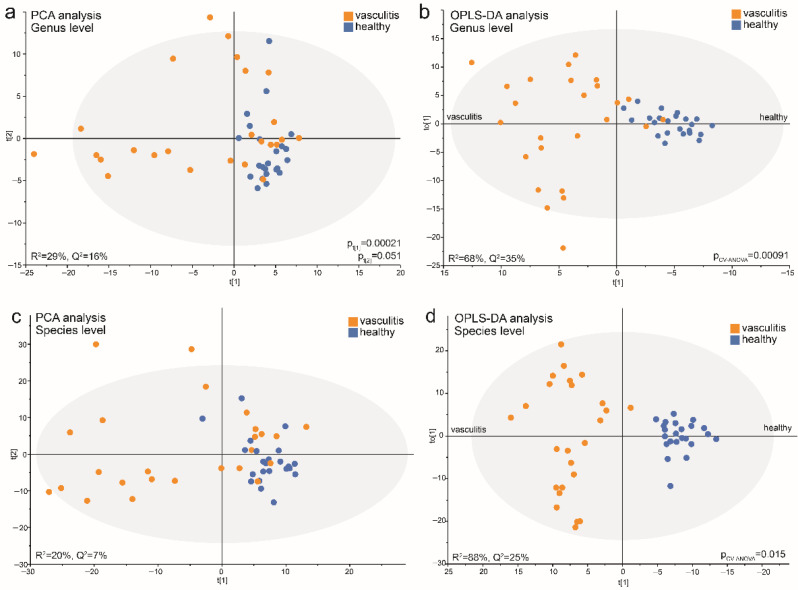
Multivariate models of saliva microbiota in acute AAV patient and control samples run at the genus and species levels. PCA score plots show the position of each participant (dot) based on microbiota composition at the (**a**) genus level and (**b**) species level in the first and second components (t[1] and t[2]). OPLS-DA loading scatterplots based on models including participant status as a dependent variable (t[1]) and abundance and prevalence of genera (**c**) or species (**d**) detected in saliva samples as the independent block (x). The subgroup intra-variation is observed in the orthogonal loading score (to[1]). The model goodness-of-fit parameter, R^2^, represents the fraction of the variance of the y variable explained by the model, whereas Q^2^ suggests the model’s predictive performance. Model validation by 7-fold CV-ANOVA is shown for models (**c**,**d**). *p*-values obtained by GLM analysis on PCA component loading scores (t[1] and t[2]) when comparing acute AAV and control samples are shown for models (**a**,**b**).

**Figure 7 microorganisms-10-01572-f007:**
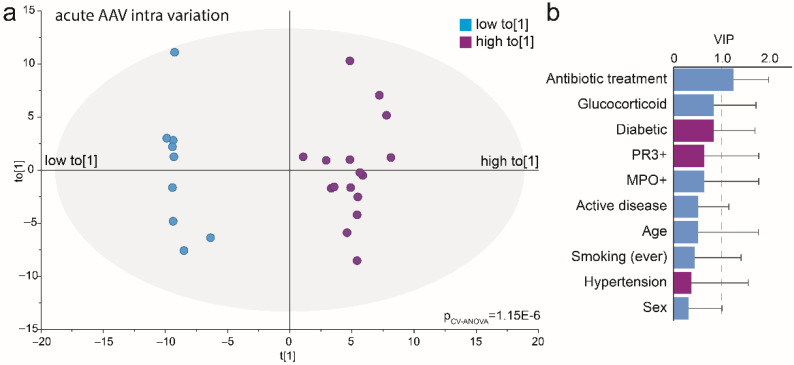
Multivariate analysis of acute AAV patient’s saliva microbiota. (**a**) Loading scatterplot based on an OPLS-DA model comparing acute AAV subgroups based on the intra-variation orthogonal loading score (to[1] shown in Figure 6a) as the dependent variable (t[1]) and clinical treatments, participant characteristics, and abundance and prevalence of detected genera in saliva samples as the independent block (x). (**b**) Bar graph of VIP values for included clinical and treatment variables. VIP: variable of importance in the projection of the dependent variable and VIP > 1.0 were defined as influential.

**Table 1 microorganisms-10-01572-t001:** Characteristics of study participants.

	Pre-AAV *n* = 85	Established AAV *n* = 78	Controls *n* = 85	Acute AAV *n* = 25	Controls *n* = 23
Women, *n* (%)	49 (57.6)	41 (52.6)	49 (57.6)	15 (60)	12 (52)
Age at diagnosis, mean (SD), years	-	53.2 (19.7)	-		
Age at sampling, mean (SD), years	52.3 (16.7)	64.3 (19.1)	52.0 (16.7)	60.9 (18.0)	61.6 (18.2)
Ever smoker, *n* (%)	10/21 (47.6)	31/74 (41.9)	8/17 (47.1)	8/22 (36.4)	7/23 (30.4)
Current smoker, *n* (%)	2/21 (9.5)	6/74 (8.1)	5/17 (29.4)	1 (4)	0
Diabetes, *n* (%)	-	13 (19.7)	-	4 (16.0)	0
Pre-dating time, mean (SD) years	4.4 (3.1)	-	-	-	-
Disease duration, mean (SD) years	-	9.7 (7.1)	-	-	
MPO-ANCA+, *n* (%)	9 (10.6)	24 (30.8)	2 (2.4)	5 (20)	-
PR3-ANCA+, *n* (%)	21 (24.7)	54 (69.2)	2 (2.4)	20 (80)	-
GPA diagnosis ^1^, *n* (%)	-	53 (67.9)	-	20 (80)	-
MPA diagnosis ^1^, *n* (%)	-	23 (29.5)	-	4 (16)	-
EGPA diagnosis ^1^, *n* (%)	-	-	-	1 (4.0)	-
Serum/plasma, *n* (%)/*n* (%)	68(80)/17(20)	-/78(100)	68(80)/17(20)	-	-
Treatment during sampling					
Corticosteroids, *n* (%)	-	42/73 (57.5)	-	7 (28.0) ^2^	-
Cytotoxic drugs, *n* (%)	-	65/73 (89.0) ^3^	-	5/25 ^4^	-
Antibiotics (long term) ^5^, *n* (%)	-	19 (27.9)	-	14 (56.0)	-

^1^ European Medicines Agency, GPA = granulomatosis with polyangiitis, MPA = Microscopic polyangiitis, EGPA = Eosinophilic granulomatosis with polyangiitis [24]. ^2^ 7 received corticosteroids before sampling, and 10 did so on the sampling day. ^3^ Cytotoxic maintenance drugs were methotrexate, azathioprine, mycophenolate mofetil, rituximab, and tacrolimus. ^4^ At sampling: 2 on methotrexate, 1 on azathioprine^,^ 1 on rituximab, and 1 on cyclophosphamide. ^5^ Antibiotic treatment within the last 3 months predating sampling.

## Data Availability

The paired-end 16S sequences presented in this study can be found at https://www.ncbi.nlm.nih.gov/PRJNA862307. The clinical datasets analyzed during the current study are not publicly available because of Swedish legislation. Still, anonymized data may be available from the corresponding author on reasonable request and adequate ethics approval. Other data are available upon reasonable request and acquired mandatory ethics approvals.

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
