# Peer review of "Oral Microbiota Profile in Patients with Anti-Neutrophil Cytoplasmic Antibody–Associated Vasculitis"

_microorganisms, 2022, doi:10.3390/microorganisms10081572_

Round 1

Reviewer 1 Report

Dear Authors, 

I have read with great interest your manuscript entitled “Oral microbiota profile in patients with anti-neutrophil cytoplasmic antibody–associated vasculitis”.
I congratulate with the Authors for the excellent work, as it is well-designed and structured. 

I just signal that you have up to 10 keywords. Please, add more keywords to be sure the manuscript could be searched properly. 

Thank you for your hard work. 

Author Response

Reviewer 1

I have read with great interest your manuscript entitled “Oral microbiota profile in patients with anti-neutrophil cytoplasmic antibody-associated vasculitis”.
I congratulate with the Authors for the excellent work, as it is well-designed and structured. 

I just signal that you have up to 10 keywords. Please, add more keywords to be sure the manuscript could be searched properly. 

Thank you for your hard work. 

Answer: Thank you. Following your advice, we have added the following key words: Anti-neutrophil cytoplasmatic antibody-associated vasculitis, Granulomatosis with polyangiitis (GPA), Microscopic polyangiitis (MPA), Proteinase 3 (PR3)-ANCA, Myeloperoxidase (MPO)-ANCA on rows 39-41.

Reviewer 2 Report

Dear Authors,

the manuscript “Oral microbiota profile in patients with anti-neutrophil cyto-2 plasmic antibody–associated vasculitis” was aimed to study and compare the immune response to oral bacteria and the oral bacterial profile in AAV before symptom onset, in acute AAV, and inactive established disease after treatment and controls.

The authors have submitted quite an interesting manuscript. The topic is interesting and the intent to know the role of oral microbiota in AAV is appreciable for a better and earlier diagnosis and therapy.

I have found a few issues that, once addressed, will improve the manuscript.

Author Response

Reviewer 2

The manuscript “Oral microbiota profile in patients with anti-neutrophil cyto-2 plasmic antibody–associated vasculitis” was aimed to study and compare the immune response to oral bacteria and the oral bacterial profile in AAV before symptom onset, in acute AAV, and inactive established disease after treatment and controls. The authors have submitted quite an interesting manuscript. The topic is interesting and the intent to know the role of oral microbiota in AAV is appreciable for a better and earlier diagnosis and therapy. I have found a few issues that, once addressed, will improve the manuscript.

Answer: We thank the reviewer for the overall assessment and would have been happy to address any issues mentioned by the reviewer but found no specification.

Reviewer 3 Report

This study analyzed the immune response to oral bacteria and the oral bacterial profile in AAV before symptom onset, in acute AAC, and inactive established disease after treatment in comparison with controls. Their findings indicated that the IgG profiles against oral bacteria differed between pre-AAV, established AAV, and controls, and microbiota profiles between acute AAV and controls. Although this is an interesting work, several concerns should be addressed as outlined below:

1. To detect the immune response to oral bacteria in pre-AAV and established AAV, IgG levels against 53 oral bacterial species/subspecies in 27 genera were screened by immunoblotting. The reasons for the selection of IgG antibody and 53 oral bacterial species in this study are unconvincing. 

2. In the part of the comparison of periodontitis severity between established AAV and controls, the authors only used deep PPD (>=6 mm) as a proxy. However, other index such as attachment loss, tooth loss, should also be taken into consideration.

3. In the first part of this study, the authors compared adaptive immune responses to oral bacteria in pre-AAV versus established AAV. However, in the second part, 25 acute AAC patients were recruited for the Illumina amplicon sequencing of saliva microbiota. The grouping method is lack of inherent continuity and logicality.

4. As the potential impact of antibiotics on oral microbiota, saliva sampling should be excluded subjects who had antibiotic administration in recent two months.

Author Response

Reviewer 3

This study analyzed the immune response to oral bacteria and the oral bacterial profile in AAV before symptom onset, in acute AAC, and inactive established disease after treatment in comparison with controls. Their findings indicated that the IgG profiles against oral bacteria differed between pre-AAV, established AAV, and controls, and microbiota profiles between acute AAV and controls. Although this is an interesting work, several concerns should be addressed as outlined below:

  1. To detect the immune response to oral bacteria in pre-AAV and established AAV, IgG levels against 53 oral bacterial species/subspecies in 27 genera were screened by immunoblotting. The reasons for the selection of IgG antibody and 53 oral bacterial species in this study are unconvincing.

Answer: The panel of bacteria selected for IgG screening was based on four main aspects, (i) species being reported as oral commensal species, (ii) species considered to have a role in the dysbiotic caries associated biofilm, (iii) species considered to have a role in the dysbiotic periodontitis associated biofilm, and (iv) Staphylococcus being reported in AAV. To clarify this, we have added the following sentence: “Briefly, 68 bacterial strains representing (i) oral commensal taxa (e.g., Corynebacterium matruchotii, Gemella haemolysans, Streptococcus sanguinis), (ii) the dysbiotic caries associated biofilm (e.g., Streptococcus mutans, Lactobacillus gasseri, Lactobacillus vaginalis), (iii) the dysbiotic periodontitis associated biofilm (e.g., Aggregatibacter actinomycetemcomitans, Filifactor alocis, Fusobacterium nucleatum, Porphyromonas gingivalis, Tannerella forsythia), or (iv) reported as associated with AAV (Staphylococcus aureus, Staphylococcus epidermidis) were grown for 48 h on Columbia-based blood agar, Rogosa, or chocolate agar plates under aerobic or anaerobic conditions at 37°C (Supplementary Table 1)” on lines 188-194.

  1. In the part of the comparison of periodontitis severity between established AAV and controls, the authors only used deep PPD (>=6 mm) as a proxy. However, other index such as attachment loss, tooth loss, should also be taken into consideration.

Answer: We agree with the reviewer that clinical attachment loss or bone loss would be optimal to consider in the estimation of periodontal status. As for tooth loss this is considered in the imputation algorithm as described previously (ref 1). Unfortunately, clinical attachment loss or bone loss are not systematically reported to the national register on caries and periodontitis in Sweden which we had to rely on since the participants were recruited from the whole country. Thus, examining them ourselves was not feasible for geographical reasons and the ethical permission did not include contacting the dentist for access to eventual x-rays. We have added the reference for the algorithm used for estimation of missing teeth in line 147, and the following comment to the text in rows 649-652 An additional weakness is that the assessment of periodontitis was based on pocket depth alone; however, clinical attachment loss or bone loss are not yet systematically reported to the national register on dental status in Sweden.

  1. In the first part of this study, the authors compared adaptive immune responses to oral bacteria in pre-AAV versus established AAV. However, in the second part, 25 acute AAC patients were recruited for the Illumina amplicon sequencing of saliva microbiota. The grouping method is lack of inherent continuity and logicality.

Answer: We acknowledge the comment made by the reviewer but regrettably none of the biobanks had saliva or any other type of sample that could be used to assess the oral microbiota before disease onset or in the established state. Therefore, the only option was to collect saliva from incident cases, i.e., new incidence or relapse with further limitation set by the very low incidence of AAV. Besides having mentioned this in the discussion in the R0 version, we have now added this clarification to line 384 too.

  1. As the potential impact of antibiotics on oral microbiota, saliva sampling should be excluded subjects who had antibiotic administration in recent two months.

Answer: We appreciate this comment and have expanded the analyses in relation to antibiotic exposure. First, we have added a sentence describing the numbers of individuals who had received antibiotics. This reads “Of the individuals who had a first acute attack of vasculitis, 9 had received antibiotics between 1-10 days at the primary care and 5 of the individuals with an acute relapse were either on long-term antibiotics or had received it during the latest 2 weeks. Thus, 11 AAV individuals and none of the controls had not received antibiotics for the latest 3 months”. This is found on lines 169-173. We did not omit the previous analyses with all 25 cases as that highlights a systematic effect of antibiotics per se which use of corticosteroids did not. As a follow up, we reran the analysis comparing the microbiota of the 11 cases without any recent use of antibiotics with that of the controls. The new results are found in a separate paragraph starting at line 516-527. These new results also called for edits in the discussion which are found in lines 558-580 and 618-620.

  1. Haworth, S.; Esberg, A.; Kuja-Halkola, R.; Lundberg, P.; Magnusson, P.K.E.; Johansson, I. Using national register data to estimate the heritability of periodontitis. J Clin Periodontol 2021, 48, 756-764, doi:10.1111/jcpe.13459.